# Inference-Time Computations for LLM Reasoning and Planning: An Evaluation Suite

## Abstract

We examine the reasoning and planning capabilities of large language models (LLMs) in solving complex tasks. Recent advances in inference-time techniques demonstrate the potential to enhance LLM reasoning without additional training by exploring intermediate steps during inference. Notably, OpenAI's latest reasoning model show promising performance through its novel use of multi-step reasoning and verification. Here, we explore how scaling inference-time techniques can improve reasoning and planning, focusing on understanding the tradeoff between computational cost and performance. To this end, we construct a comprehensive *evaluation suite*, known as *Sys2Bench*, and perform extensive experiments evaluating existing inference-time techniques on eleven diverse tasks across five categories, including arithmetic reasoning, logical-reasoning, common-sense reasoning, algorithmic reasoning, and planning. *Sys2Bench* provides a unified framework for revealing the strengths and limitations of current inference-time methods, setting the stage for more principled and scalable approaches to LLM reasoning.

## 1 Introduction

Large language models (LLMs) (Brown et al., 2020) have demonstrated exceptional performance across a range of natural language processing (NLP) tasks, including question answering, machine translation, sentiment analysis, and text summarization (Devlin et al., 2019; Vaswani et al., 2017). Beyond NLP, LLMs have also been adapted for multimodal tasks involving vision (Parashar et al., 2024; Lin et al., 2025) and audio (Wu et al., 2024). Building on their success in these diverse domains, researchers are increasingly using LLMs as AI agents (Deng et al., 2024; Wang et al., 2024) for complex tasks, such as robotics (Liu et al., 2023) and scientific discovery (Wang et al., 2024). These tasks require the reasoning and planning capabilities of LLMs, extending beyond simpler text comprehension.

Reasoning and planning in LLMs refer to their ability to solve complex problems by understanding, processing, and generating solutions across various domains (Hao et al., 2024). These capabilities can be analyzed from multiple perspectives; we propose a classification that organizes reasoning and planning tasks into five categories, namely arithmetic, logical, commonsense, algorithmic, and plan generation challenges. Recent advances in inference-time techniques demonstrate the potential to enhance LLM reasoning and planning without additional training. These techniques focus on decomposing complex problems into simpler intermediate steps during inference. For instance, Chain-of-Thought (Wei et al., 2022) encourages step-by-step reasoning, while Tree-of-Thought (Yao et al., 2024) chooses optimal reasoning paths using tree search. *Recently, Large Reasoning Models (LRMs) (Valmeekam et al., 2024) such as DeepSeek-R1 (DeepSeek-AI et al., 2025) and Kimi (Team et al., 2025) have achieved impressive performance by using reinforcement learning to natively train the model to "think" before answering. Yet, for standard LLMs, inference-time search/prompting techniques remain the most popular method for complex reasoning. Despite their widespread adoption, these inference-time techniques have not been uniformly evaluated across diverse planning and reasoning domains. Furthermore, as LRMs internalize step-by-step reasoning, it remains an open question whether external inference-time search algorithms can successfully transfer to, and further enhance, these natively reasoning models.*

*To address these gaps, we introduce Sys2Bench, a unified evaluation suite designed to systematically analyze inference-time techniques. Prior evaluations of these methods have largely been confined to isolated datasets or specific models. To establish a rigorous and standardized comparative framework, Sys2Bench evaluates four widely adopted inference-time strategies across eleven distinct datasets and seven diverse LLMs and LRMs. Our findings reveal that while these techniques yield notable improvements in certain contexts, their effectiveness remains highly inconsistent across domains and is frequently bottlenecked by computational costs. Ultimately, these results underscore the necessity of moving beyond unconstrained inference scaling toward the development of more principled, adaptive strategies for advancing the reasoning capabilities of foundational models.*

## 2 Related Work

**LLM Reasoning** is the ability of LLMs to logically process information and draw coherent conclusions, enabling them to solve complex problems (Saparov and He, 2023). The success of LLMs in Natural Language Generation (Radford et al., 2018) and Natural Language Understanding (Vaswani et al., 2017; Devlin et al., 2019). *A range of benchmarks have* been introduced to evaluate reasoning, covering tasks in arithmetic (Ling et al., 2017; Cobbe et al., 2021), logic (Chollet, 2019; Wang et al., 2022), common sense (Yang et al., 2018; Geva et al., 2021), and algorithmic reasoning (Yao et al., 2024). We introduce these tasks in more detail in Section 3.

**LLM Planning** involves constructing a sequence of actions to achieve defined goals (Valmeekam et al., 2023; Zheng et al., 2024). LLMs have been employed as planners or high-level controllers for robotic tasks Liu et al. (2023); Huang et al. (2022) and as agents for web navigation (Deng et al., 2024), scientific discovery (Wang et al., 2024), and autonomous vehicles (Yang et al., 2023). Despite their broad adoption, *recent work has shown that LLMs frequently fail to produce valid plans for* complex tasks (Kambhampati et al., 2024; Xie et al., 2024).

**Inference-Time Techniques** for LLMs are methods applied during output generation to improve performance *and* alignment with downstream tasks (Welleck et al., 2024). These techniques aid reasoning and planning by breaking complex tasks into smaller, manageable steps for systematic problem-solving. For instance, Chain-of-Thought prompting (CoT) (Wei et al., 2022) and its variants (Zhou et al., 2023; Kojima et al., 2022) decompose problems into sequential steps, while self-consistency (Wang et al., 2023) refines CoT by aggregating multiple responses through voting. Tree of Thought (Yao et al., 2024), Graph of Thought (Besta et al., 2024), and Monte Carlo Tree Search (Hao et al., 2023; Zhou et al., 2024) enhance problem-solving by systematically exploring reasoning paths. *Details on inference-time methods are in Section 4, with results in Section 5.*

*Large Reasoning Models (LRMs) extend inference-time techniques by internalising the search process through training with reinforcement learning (RL) (Shao et al., 2024) on model rollouts (OpenAI, 2024; DeepSeek-AI et al., 2025; Team et al., 2025). Unlike classical inference-time methods which are training free, LRMs internalize the search process during post-training. This approach yields superior reasoning performance, though it necessitates significantly longer output sequences (Muennighoff et al., 2025; Zeng et al., 2025). Building on LRM success, new RL methods (Parashar et al., 2026; Yu et al., 2025) have been developed to enhance reasoning. Despite their success, recent works have pointed to common-sense reasoning failures (Shojaee et al., 2025), or repetition of patterns in the reasoning process of LRMs (Song et al., 2026; Yan et al., 2025). Along these lines, we evaluate the effect of inference time search techniques on LRMs, as we do for traditional non-reasoning LLMs*

*We detail all evaluation tasks in Section 3 and report results in Section 5.*

## 3 Sys2Bench Problems and Datasets

In this section, we introduce **Sys2Bench**, *an evaluation suite* designed to systematically evaluate the reasoning and planning capabilities of Large Language Models (LLMs) across diverse tasks. The name Sys2Bench reflects its focus on evaluating Systematic *(System-2)* Reasoning and Planning, providing a structured framework for assessing inference-time techniques.

Table 1: Summary of the 11 datasets included in Sys2Bench.

| | Algorithmic Reasoning | | Planning | | | |
|---|---|---|---|---|---|---|
| | Game of 24 | Binpacking | Blocksworld | Trip Plan | Calendar Plan | Rubik's Cube |
| Task | Propose an arithmetic expression to reach 24. | Pack items into the fewest bins. | Plan actions to transform blocks from initial to goal state. | Plan a trip across cities for a set number of days. | Schedule a meeting considering time constraints of people. | Unscramble a scrambled 2×2 Rubik's Cube. |
| Input | A list of 4 numbers. | List of item weights and bin capacity. | Initial state of blocks and goal state. | Cities, days per city, total days, and possible flights. | Calendars with meetings and time constraints. | A scrambled 2×2 Rubik's Cube. |
| Output | An arithmetic expression. | Final list with items arranged in bins. | A sequence of actions as the plan. | A trip itinerary. | A meeting time fitting all schedules. | A sequence of rotations that unscramble the cube. |
| | Arithmetic Reasoning | | Logical Reasoning | Common Sense Reasoning | | |
| | GSM8K | AQuA | ProntoQA | StrategyQA | HotPotQA | |
| Task | Solve high school arithmetic problems. | Solve algebraic problems. | Draw a logical conclusion from a set of predicates. | Answer general knowledge questions. | Answer general knowledge questions using provided facts. | |
| Input | Arithmetic problem description. | Algebraic problem description. | A clause to verify as true or false using logical predicates. | A yes/no question. | General knowledge question with supporting facts. | |
| Output | A numerical value. | A multiple-choice option. | True or False, with reasoning. | Yes or No. | Short answer of 1 or 2 words. | |

A key motivation for this *suite* is to *rigorously examine the efficacy of scaling inference-time search*, *investigating whether increased computational overhead consistently translates to* better reasoning or problem-solving abilities. While inference-time techniques have gained traction in improving LLM performance, *current evidence suggests that* no single approach consistently *dominates* others across all *problem domains*. Thus, we argue that a more *nuanced, context-aware* exploration of reasoning strategies is essential. Sys2Bench facilitates this by *evaluating* LLMs on eleven datasets, categorized into five primary reasoning types: Arithmetic Reasoning, Logical Reasoning, Common Sense Reasoning, Algorithmic Reasoning, and Planning (summarized in Table 1).

*Additionally, with the advent of large reasoning models (OpenAI, 2024; Yang et al., 2025), it remains an open question whether human-designed search strategies provide any benefit over the model's own reasoning process learned through post-training. In Sys2Bench, we empirically investigate this question.*

## 3.1 Arithmetic Reasoning

The ability of Large Language Models (LLMs) to solve multi-step arithmetic problems remains an active area of research Snell et al. (2024); Kumar et al. (2024); Hendrycks et al. (2021). Additionally, OpenAI's o1 models (OpenAI, 2024) have prompted the research community to explore inference-time techniques to improve the arithmetic reasoning of LLMs (Zhao et al., 2024). We evaluate the arithmetic reasoning of LLMs, on **GSM8K** (Cobbe et al., 2021) and **AQuA** (Ling et al., 2017) benchmark.

**GSM8K** is a popular dataset of high-quality, linguistically diverse elementary school math word problems, designed to evaluate multi-step arithmetic reasoning. The problems typically require 2 to 8 steps of arithmetic operations, testing the ability of LLMs to perform logical deduction and basic calculations.

**AQuA** is a dataset of around 100,000 algebraic word problems with multiple-choice answers and detailed rationales. It is designed to evaluate the arithmetic reasoning and problem-solving capabilities of models, making it a challenging benchmark for LLMs.

## 3.2 Logical Reasoning

Logical reasoning involves deriving conclusions based on a structured sequence of rules, or premises. The evaluation of the ability to reason logically by LLM helps assess their ability to solve structured and complex decision-making problems (Chollet, 2019). We use **ProntoQA** (Saparov and He, 2023) to evaluate the logical reasoning ability of LLMs.

**ProntoQA** is a dataset developed to evaluate an LLM's ability to reason and generate explicit reasoning chains for first-order logic-based queries (Barwise, 1977). It challenges models to not only produce correct answers but also provide detailed, step-by-step reasoning paths that justify their conclusions.

### 3.3 Common Sense Reasoning

Common Sense Reasoning is the process of drawing conclusions from implicit everyday knowledge. Evaluating this skill ensures that LLMs provide accurate and contextually appropriate responses. We evaluate this type of reasoning using the **StrategyQA** Geva et al. (2021) and **HotPotQA** Yang et al. (2018) datasets.

**StrategyQA** is a benchmark designed to assess a model's ability to perform implicit multi-step reasoning using general knowledge or common sense facts. It consists of yes/no questions where the goal is to arrive at the correct answer by generating and verifying intermediate reasoning steps.

**HotPotQA** is a large-scale dataset designed to evaluate how effectively models combine information from multiple documents to answer general knowledge questions. It features diverse question types and tests the use of sentence-level evidence for accurate and explainable multi-hop reasoning.

### 3.4 Algorithmic Reasoning

We apply LLMs to tackle complex NP-hard and NP-complete tasks, requiring them to evaluate constraints and propose optimized algorithms that achieve practical solutions. Such problems assess the application of LLMs to combinatorial optimization and resource allocation tasks (Liu et al., 2024; Romera-Paredes et al., 2024). We use **Game of 24** (Yao et al., 2024), and **Bin Packing** (Liu et al., 2024; Romera-Paredes et al., 2024).

**Game of 24** is a dataset where the goal is to form an equation evaluating to 24 using '+', '-', '*', or '/' with a list of four numbers. As an NP-complete problem with multiple solutions, it challenges an LLM to efficiently generate expressions by focusing only on operations that can lead to the target value.

**Bin Packing** is a task where the goal is to find the least number of bins needed to pack a list of items. Specifically, a list of $N$ items of weight $[W_1, W_2, ...W_n]$ is given, which must be divided into bins $B_1, B_2, B_3...B_m$. The sum of weights in each bin must not exceed the bin capacity $C$, and the objective is to minimize the total number of bins $m$. Formally, the task can be written as:

$$\min \quad m \quad \text{subject to} \quad \begin{cases} \bigcup_{j=1}^{m} B_j = \{1, \ldots, n\}, \quad B_j \cap B_{j'} = \varnothing \quad (\forall j \neq j'), \\ \\ \sum_{i \in B_j} W_i \leq C \quad (\forall j). \end{cases} \tag{1}$$

### 3.5 Planning

A planning problem is defined by $(S_0, A, G)$, where $S_0$ stands for an initial state, $A$ is the set of actions needed to achieve the goal $G$. Planning problems require LLMs to demonstrate multistep reasoning, and sound decision making to arrive at correct solutions. These problems have broad applications in robotics and agent-based systems. Our evaluation focuses on four planning problems: BlocksWorld (Valmeekam et al., 2023), Rubik's Cube (Ding et al., 2024), TripPlan, and CalendarPlan (Zheng et al., 2024).

**BlocksWorld** is a popular dataset to evaluate the planning capabilities of LLMs. Each task involves transitioning from an initial block configuration to a target configuration, which requires LLMs to generate a sequence of actions to achieve the goal.

**Rubik's Cube** requires an LLM to solve a scrambled $2 \times 2$ cube by restoring each face to a uniform color. Starting from a scrambled cube, the LLM must generate a valid plan of cube rotations to achieve the goal.

**Trip Plan** challenges an LLM to plan a travel itinerary that satisfies constraints on cities, dates, and flight connectivity, ensuring that all cities are visited as specified.

**Calendar Plan** is a dataset designed to schedule a meeting by aligning the availability of a group of people. The goal is to find a feasible time slot that accommodates all the constraints of the participants.

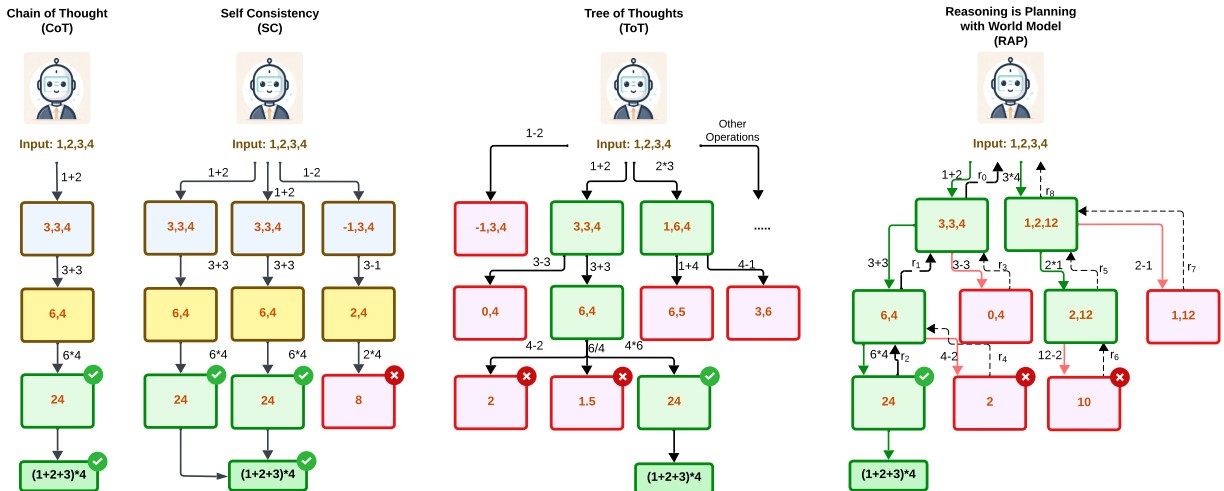

Figure 1: Overview of Inference-Time Techniques evaluated on the Game of 24 dataset. We evaluate four inference-time reasoning techniques. Chain of Thought (CoT) (Wei et al., 2022) solves problems through a linear sequence of reasoning steps. Self-Consistency (SC) (Wang et al., 2023) extends CoT by selecting answers through majority voting over multiple reasoning chains. Tree of Thoughts (ToT) (Yao et al., 2024) uses tree search to explore and expand reasoning paths. Reasoning as Planning (RAP) Hao et al. (2023) combines Monte Carlo Tree Search (MCTS) with the LLM as a world model to reward reasoning steps and guide tree growth toward the answer.

## 4 Sys2Bench Baseline Methods

*In Sys2Bench, we evaluate popular inference-time techniques commonly used to enhance the complex reasoning capabilities of LLMs, broadly referred to as System 2 abilities (Wei et al., 2022).* While these techniques have typically been applied to specific tasks, we analyze their performance comprehensively in Sys2Bench. Sys2Bench allows us to uncover patterns and limitations that may not be previously evident. We summarize these methods in Fig. 1.

**Chain of Thought** (CoT) enables LLMs to solve complex problems by breaking them into intermediate reasoning steps, improving their logical coherence and accuracy Wei et al. (2022). CoT enhances structured problem-solving of LLMs by providing in-context examples of step-by-step reasoning during inference.

**Self Consistency** (SC) extends CoT by generating multiple reasoning paths for a problem and selecting the most consistent answer through majority voting (Wang et al., 2023).

**Tree of Thoughts** (ToT) uses structured tree search to enhance reasoning in LLMs by systematically exploring multiple paths, with the LLM evaluating its own intermediate generations to decide which paths to expand (Yao et al., 2024). Evaluation can be performed by rating LLM generation on a scale of 1-10 or using logits for scoring.

ToT has three search strategies: depth-first search (DFS), breadth-first search (BFS), and beam search. In our experiments, we use beam search because it performs the best amongst all variants.

**Reasoning as Planning with World Models** (RAP) reformulates reasoning as a planning problem, where the LLM acts as both the reasoning agent and the world model (Hao et al., 2023). The reasoning agent generates potential reasoning paths, while the world model simulates and evaluates these paths. Specifically, RAP uses Monte Carlo Tree Search (MCTS) (Coulom, 2006) to explore and refine reasoning paths.

Unlike ToT, which does exhaustive tree search, RAP dynamically prioritizes high-potential paths using MCTS, resulting in improved performance. RAP requires logits for MCTS, which is why it is exclusively implemented on LLaMA. Since RAP requires extensive prompt engineering to frame all tasks as planning

problems, we evaluate it on a subset of tasks, including GSM8K, AQuA, ProntoQA, StrategyQA, Game of 24, Binpacking, Blocksworld, and Rubik's Cube.

## 5 Experiments

In this section, we present the experiments conducted on various tasks in the Sys2Bench evaluation suite. We begin by outlining the experimental setup, detailing the models and implementation specifics of the inference-time methods. Next, the results for the different inference-time methods are shown in Table 2 and Table 3.

### 5.1 Setup

In this subsection, we provide details about the experimental setup used to evaluate the performance of various inference-time techniques in Sys2Bench. We describe the models, the implementation specifics of the inference-time methods, and the metrics used for evaluation.

**Models** evaluated in Sys2Bench, consist of three LLaMa 3.1 models, two GPT-based models, and two large reasoning models (LRMs). The LLaMa 3.1 variants are 8B, 70B, and 405B, while the GPT-based models include GPT-4o and GPT-4o-mini. Additionally, the O1 and O1-mini models are tested as part of our LRM evaluation. By default, we use a temperature of 0.8 across all models for generation.

**Chain of Thought (CoT)** involves including in-context learning examples in the prompt. In Sys2bench we limit this to five examples per prompt. These examples are selected from the in-context examples provided by the dataset or the training set. If neither is available, we use a subset of test examples and evaluate the remaining test instances.

**Self Consistency (SC)** follows the same settings as CoT. We generate five CoT responses from the LLM and determine the final output through majority voting.

**Tree of Thought (ToT)** implementation in Sys2Bench uses beam search. The beam size is 5 for most tasks except planning tasks, where the number of possible actions at each state is larger, thus, the beam size is increased to 10. By default, beam ranking is performed by asking the LLM to rate outputs on a scale of 1 to 10, except for LLaMa 3.1 8B, where logits are used instead. Finally, the search depth is task-dependent, ranging from 4 for Game of 24 to 20 for Trip Plan.

**Reasoning as Planning (RAP)** uses Monte Carlo Tree Search (MCTS) with up to 10 rollouts during inference. Due to its reliance on a reward model that requires logits, RAP is implemented exclusively on LLaMa 3.1 8B and *70B due to hardware constraints*. Similar to ToT, the search depth in RAP varies depending on the task.

**Input Output Prompting (IO)** is utilized with LRMs, as they generate their own reasoning steps and do not require in-context learning examples. Instead, we provide the necessary format and instruct the models to respond in the same format.

**Metric** used across all tasks is accuracy. Note that the context of accuracy differs depending each task. For arithmetic, commonsense, and algorithmic reasoning tasks, accuracy is measured on the correctness of the final answer. Logical reasoning tasks, namely, ProntoQA, accuracy measures the ability of an LLM to generate the correct reasoning chain. Finally, for planning tasks, accuracy measures the correctness of the proposed plan.

### 5.2 Results

In this subsection, we present the results of *our study*, organized by the types of reasoning outlined in Section 3. This grouping allows for a clearer comparison of performance across tasks, demonstrating the strengths and limitations of different inference-time techniques.

**Arithmetic Reasoning** tasks in Table 2 have strong results with CoT. Performance further improves with SC, as it reduces the impact of randomness in the CoT answers. However, this strong performance

does not transfer to tree search methods. ToT significantly underperforms on this task, as its approach of prompting the LLM to explore multiple reasoning paths relies on the LLM generating and selecting correct intermediate reasoning steps. Since LLMs struggle with self-verification (Huang et al., 2024), it selects incorrect intermediate arithmetic steps, leading to wrong answers. In contrast, RAP shows modest gains on the GSM8K dataset, benefiting from the LLM's role as a world model to select better arithmetic steps. However, RAP still underperforms SC on AQuA, indicating that tree search methods are not well-suited for arithmetic reasoning tasks. Meanwhile, LRMs deliver exceptional arithmetic reasoning performance, as shown in Table 3, highlighting their strength in arithmetic.

**Logical Reasoning** results in Table 2 show interesting trends. For instance, SC improves performance over CoT on LLaMa 3.1 8B and 70B. However, for LLaMa 3.1 405B and GPT-based models, SC results in performance drops, as it increases the likelihood of generating multiple incorrect reasoning chains in the ProntoQA task, where evaluation focuses on the accuracy of these chains. Majority voting does not help when the LLM outputs multiple wrong reasoning chains. Consistent with arithmetic reasoning, tree search methods such as ToT and RAP also underperform in this task, indicating their limitations in logical reasoning. Finally, as shown in Table 3, LRMs do not consistently outperform LLMs on this task, with O1 performing worse than GPT-4o on this task.

**Common Sense Reasoning** performance of CoT and SC improves with increasing LLM size. However, tree search methods show unique trends. Specifically, both RAP and ToT generate supporting facts for each question, but their effectiveness varies by task. To be specific, in StrategyQA, the binary output (yes or no) enables LLaMA models to effectively utilize the generated facts, leading to improved performance. In contrast, for HotPotQA, tree search is not effective as the LLM needs to output short answers. Additional facts often cause LLM hallucinations and increased error rates. Furthermore, compared to other tasks, performance improvements seen with LRMs are limited (see Table 3).

**Algorithmic Reasoning** tasks include the Game of 24 and Binpacking datasets, as described in Section 3. Table 2 shows that both CoT and SC underperform on these tasks. Due to the combinatorial optimization nature of these tasks, that require extensive search, tree search methods perform well on all models, except LLaMa 3.1 8B. The smaller size of LLaMa 3.1 8B limits the model to accurately evaluate and determine the next steps toward a solution. When comparing LLMs to LRMs, results in Table 3 highlight the potential of O1-mini and O1 in solving NP-Hard and NP-Complete problems, with O1 slightly underperforming O1-mini on Game of 24.

**Planning** tasks are the most challenging in Sys2Bench. Generally, CoT and SC performance improves with larger model sizes, and SC consistently outperforming CoT.

Tree search methods show mixed results across tasks and models. On smaller models, such as LLaMa 3.1 8B and GPT-4o-mini, ToT shows improvements on tasks like Blocksworld and TripPlan. However, for larger models and other tasks, ToT often decreases performance. This is because planning tasks require LLMs to generate actions to solve problems, and incorrect actions can lead to incorrect solutions. Although ToT is intended to help LLMs explore multiple reasoning paths, which in planning means considering different actions, LLMs often fail to generate accurate actions, ultimately reducing performance. The other tree search method, RAP, performs exceptionally well on Blocksworld by leveraging the LLM as a world model to predict future states and rewards.

Compared to LLMs, LRMs perform significantly better on planning tasks, with O1 achieving near-perfect results on Blocksworld. However, the Rubik's Cube task remains challenging for all methods and models, as it requires advanced spatial reasoning and precise prediction of the consequences of each action. Both LLMs and LRMs currently lack the reasoning capabilities needed for this task, making it out-of-distribution (OOD) for current language models.

### 5.3 Insights

We extend our main experiments to provide additional insights and uncover important trends. As the research community shows increasing interest in inference-time techniques and improving LLM reasoning, these findings offer valuable contributions to ongoing discussions.

Table 2: Results of Inference Time Techniques across diverse tasks show that as model size increases, performance of CoT (CoT) (Wei et al., 2022) and Self Consistency (SC) (Wang et al., 2023) improves. However, this trend doesn't extend to tree search methods like Tree of Thought (ToT) (Yao et al., 2024), where performance does not improve with the bigger models. Furthermore, a comparison between ToT and Reasoning as Planning with World Models (RAP) (Hao et al., 2023) shows that RAP outperforms ToT in planning and arithmetic reasoning tasks but lags in commonsense reasoning while performing equally in algorithmic reasoning tasks. All methods and LLMs fail to solve the Rubik's Cube planning task. This failure can be attributed to the spatial understanding capabilities required for the task, which are currently out of distribution (OOD) for existing LLMs.

| Methods | Algorithmic Reasoning | | | | | | | | | | Logical Reasoning | | | | |
| | GSM8K | | | | | AQuA | | | | | ProntoQA | | | | |
| | LLaMa 3.1 | | | GPT | | LLaMa 3.1 | | | GPT | | LLaMa 3.1 | | | GPT | |
| | 8B | 70B | 405B | 4o mini | 4o | 8B | 70B | 405B | 4o mini | 4o | 8B | 70B | 405B | 4o mini | 4o |
|---|---|---|---|---|---|---|---|---|---|---|---|---|---|---|---|
| Chain of Thought Methods | | | | | | | | | | | | | | | |
| *Zero Shot* | 65.2 | 85.7 | 87.3 | 73.8 | 84.9 | 38.3 | 56.8 | 68.0 | 59.6 | 71.1 | 15.3 | 42.6 | 71.1 | 41.4 | 71.8 |
| CoT | 79.8 | 95.5 | 97.0 | 92.6 | 94.7 | 58.7 | 77.2 | 78.0 | 73.6 | 79.9 | 45.8 | 82.6 | 91.0 | 61.4 | 91.8 |
| SC @ 5 | 86.7 | 96.5 | 97.5 | 93.3 | 94.9 | 70.9 | 85.8 | 86.2 | 79.9 | 83.9 | 54.2 | 88.4 | 89.0 | 58.0 | 91.4 |
| Tree Search Methods | | | | | | | | | | | | | | | |
| ToT | 60.0 | 91.5 | 96.0 | 91.5 | 93.5 | 44.8 | 78.0 | 85.8 | 81.1 | 78.0 | 13.5 | 24.2 | 62.6 | 42.0 | 32.8 |
| RAP | 87.3 | 90.1 | – | – | – | 68.1 | – | – | – | – | 0.0 | 32.1 | – | – | – |
| Common Sense Reasoning | | | | | | | | | | | Algorithmic Reasoning | | | | |
| | HotPotQA | | | | | StrategyQA | | | | | Game of 24 | | | | |
| | LLaMa 3.1 | | | GPT | | LLaMa 3.1 | | | GPT | | LLaMa 3.1 | | | GPT | |
| | 8B | 70B | 405B | 4o mini | 4o | 8B | 70B | 405B | 4o mini | 4o | 8B | 70B | 405B | 4o mini | 4o |
| Chain of Thought Methods | | | | | | | | | | | | | | | |
| *Zero-Shot* | 4.4 | 25.6 | 33.1 | 25.2 | 44.7 | 26.3 | 41.2 | 56.2 | 46.2 | 59.4 | 0.0 | 1.0 | 1.0 | 3.0 | 10.0 |
| CoT | 13.8 | 30.6 | 41.0 | 38.6 | 52.8 | 46.0 | 61.5 | 76.0 | 76.6 | 79.2 | 6.0 | 8.0 | 7.0 | 13.0 | 14.0 |
| SC @ 5 | 20.6 | 36.6 | 45.6 | 40.6 | 52.6 | 53.5 | 66.0 | 78.5 | 76.0 | 79.8 | 6.0 | 8.0 | 6.0 | 15.0 | 18.0 |
| Tree Search Methods | | | | | | | | | | | | | | | |
| ToT | 23.0 | 30.0 | 31.5 | 31.4 | 38.2 | 68.0 | 82.0 | 79.5 | 67.5 | 73.5 | 1.0 | 59.0 | 69.0 | 42.0 | 62.0 |
| RAP | – | – | – | – | – | 58.5 | 78.9 | – | – | – | 1.0 | 63.0 | – | – | – |

*(Algorithmic Reasoning — Binpacking columns continued)*

| Methods | Binpacking | | | | |
| | LLaMA 3.1 | | | GPT | |
| | 8B | 70B | 405B | 4o mini | 4o |
|---|---|---|---|---|---|
| *Zero-Shot* | 0.0 | 23.0 | 30.0 | 11.0 | 45.0 |
| CoT | 6.0 | 33.0 | 45.0 | 31.0 | 75.0 |
| SC @ 5 | 6.0 | 45.0 | 64.0 | 41.0 | 86.0 |
| ToT | 1.0 | 46.0 | 81.0 | 53.0 | 77.0 |
| RAP | 1.0 | 44.0 | – | – | – |

| Methods | Planning | | | | | | | | | | | | | | |
| | Blocksworld | | | | | Trip Plan | | | | | Calendar Plan | | | | |
| | LLaMa 3.1 | | | GPT | | LLaMa 3.1 | | | GPT | | LLaMa 3.1 | | | GPT | |
| | 8B | 70B | 405B | 4o mini | 4o | 8B | 70B | 405B | 4o mini | 4o | 8B | 70B | 405B | 4o mini | 4o |
|---|---|---|---|---|---|---|---|---|---|---|---|---|---|---|---|
| Chain of Thought Methods | | | | | | | | | | | | | | | |
| *Zero-Shot* | 0.0 | 0.0 | 1.1 | 0.0 | 4.2 | 1.2 | 16.2 | 22.2 | 20.4 | 31.6 | 0.0 | 15.6 | 29.2 | 4.2 | 18.8 |
| CoT | 3.5 | 26.1 | 48.7 | 18.4 | 37.5 | 12.3 | 29.5 | 27.0 | 5.3 | 6.3 | 10.4 | 31.2 | 44.8 | 26.0 | 47.0 |
| SC @ 5 | 4.5 | 30.7 | 52.1 | 21.2 | 41.5 | 12.0 | 32.3 | 34.3 | 5.0 | 5.8 | 11.6 | 38.0 | 45.6 | 29.6 | 47.4 |
| Tree Search Methods | | | | | | | | | | | | | | | |
| ToT | 13.9 | 4.6 | 19.9 | 23.1 | 12.4 | 2.0 | 32.5 | 29.5 | 7.8 | 19.5 | 16.8 | 32.0 | 40.0 | 29.0 | 41.4 |
| RAP | 46.8 | 50.5 | – | – | – | – | – | – | – | – | – | – | – | – | – |

*(Planning — Rubik's Cube columns continued)*

| Methods | Rubik's Cube | | | | |
| | LLaMa 3.1 | | | GPT | |
| | 8B | 70B | 405B | 4o mini | 4o |
|---|---|---|---|---|---|
| *Zero-Shot* | 0.0 | 0.0 | 0.0 | 0.0 | 0.0 |
| CoT | 0.6 | 0.0 | 0.0 | 0.6 | 0.0 |
| SC @ 5 | 0.0 | 0.6 | 0.6 | 0.6 | 0.6 |
| ToT | 0.6 | 0.6 | 0.6 | 0.0 | 0.6 |
| RAP | 0.6 | 0.0 | – | – | – |

**Inference-time compute scaling is limited by LLM bias**. These techniques aim to improve LLM reasoning by guiding them to generate intermediate steps, simplifying complex tasks into smaller, manageable parts. However, this premise is flawed as LLMs do not exhaustively search for all reasoning paths and remain biased toward certain ones. As inference-time compute scales, this bias persists, limiting exploration and leading to diminished performance. As task complexity increases, this issue becomes worse, exacerbating errors in reasoning and decision-making.

Our Sys2Bench experiments show this trend in arithmetic and logical reasoning. In these tasks, LLMs excel with CoT but struggle with tree search, failing to explore reasoning paths and select the correct one.

**Tree search struggles with increasing complexity, performing significantly worse than CoT.** As shown in Fig 2, its benefits diminish beyond a depth of 4 for the TripPlanning and Blocksworld tasks on LLaMa 3.1 405B. Note that, LLaMa 3.1 405B has a strong CoT performance in challenging planning tasks

Table 3: Results of large reasoning models (LRMs). We report the results of IO prompting on LRMs, including OpenAI O1-mini and O1 (OpenAI, 2024). Overall, LRMs achieve state-of-the-art performance, with O1 outperforming O1-mini on all tasks except the Game of 24. Similar to LLMs, LRMs also struggle with the Rubik's Cube task, indicating a lack of spatial understanding.

| | Arithmetic Reasoning | | Logical Reasoning | Common Sense Reasoning | | Algorithmic Reasoning | | Planning | | | |
|---|---|---|---|---|---|---|---|---|---|---|---|
| | GSM8K | AQuA | ProntoQA | HotPotQA | StrategyQA | Game of 24 | Binpacking | Blocksworld | Trip Plan | Calendar Plan | Rubik's Cube |
| O1 Mini | 98.0 | 92.0 | 64.0 | 35.0 | 74.0 | 77.0 | 90.0 | 48.3 | 24.0 | 88.2 | 0.0 |
| O1 | 98.0 | 91.0 | 74.0 | 59.0 | 81.0 | 73.0 | 99.0 | 99.2 | 58.3 | 90.0 | 0.6 |

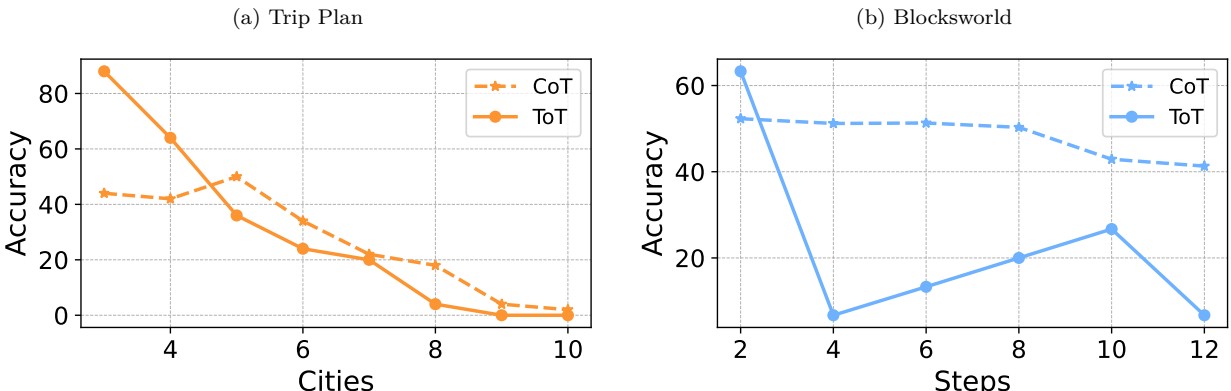

(a) Trip Plan  (b) Blocksworld

Figure 2: ToT performance declines as task complexity increases. In (a) Trip Plan and (b) Blocksworld, the number of steps or cities represents the required tree depth for LLM inference-time search. While ToT performs well for smaller depths, performance deteriorates with problem complexity. Notably, CoT achieves better performance with significantly lower computational resources, as shown in Table 5. These results are on the LLaMa 405B model.

and ideally ToT should lead to further improvements. However, as complexity grows, generating the right intermediate steps becomes crucial, leading to worse performance of ToT compared to CoT. A potential explanation for this observation is the inherent bias of LLMs at each step of the reasoning process. These biases may propagate through successive steps, leading to cumulative errors that degrade ToT performance.

*Rubik's Cube Reveals Limitations of Reasoning Models in Combinatorial Problem Solving*
*Rubik's Cube is a challenging benchmark that pushes even the strongest reasoning models to their limits, as solving it requires understanding the vast combinatorial space of possible states and move sequences. Human solvers have developed well-known algorithms built around powerful move patterns, such as OLL and PLL, that dramatically reduce this complexity. However, these moves are only valid under specific cube configurations and demand precise situational awareness to apply correctly. LRMs appear to recognize these algorithmic patterns from pretraining data Yan et al. (2025), but struggle to determine when the conditions for applying them are actually met. This leads to a characteristic failure mode where the model attempts to invoke a memorized solution strategy in the wrong context, producing moves that are locally familiar but globally invalid Shojaee et al. (2025); Song et al. (2026).*

As shown in Fig. 3, we compare GPT-4o, O1-Mini, and O1 based on how frequently their generated moves in the Rubik's Cube task align with known online algorithms. Our analysis shows that GPT-4o and O1-Mini repeat these popular moves in nearly all cases, with rates of 75% and 90%, respectively. While O1 performs better, it still follows common move sequences about 20% of the time.

**There is a tradeoff between performance and the cost of inference-time methods**. Table 5 includes results across Blocksworld, Game of 24, and GSM8K using LLaMa 3.1 8B with the number of tokens generated. Compared to CoT and SC, ToT and RAP have significantly higher computational costs and increased token usage does not yield better performance. Additionally, solving 100 Game of 24 problems with GPT-4o and ToT costs around $60 due to high token usage, with costs rising for larger models and

Table 4: *Results on Qwen3 models across inference-time search methods.*

| Method | Game of 24 | Blocksworld | Rubik's Cube | AQuA | GSM8K | Binpacking |
|---|---|---|---|---|---|---|
| | | | *Qwen3-8B* | | | |
| Zero-Shot | 3.0 | 0.0 | 0.0 | 33.3 | 86.7 | 0.0 |
| CoT | 36.0 | 12.1 | 0.0 | 78.2 | 92.9 | 26.0 |
| ToT | 47.0 | 16.2 | 0.0 | 62.1 | 88.9 | 37.0 |
| RAP | 43.0 | 27.3 | 0.0 | 59.3 | 83.2 | 41.0 |
| Thinking | 55.0 | 29.7 | 1.0 | 83.1 | 93.7 | 33.0 |
| | | | *Qwen3-32B* | | | |
| Zero-Shot | 5.0 | 2.3 | 0.0 | 44.8 | 87.7 | 3.0 |
| CoT | 46.0 | 18.5 | 0.0 | 80.2 | 93.7 | 49.0 |
| ToT | 61.0 | 26.2 | 1.0 | 75.7 | 91.1 | 72.0 |
| RAP | 59.0 | 67.1 | 0.0 | 73.1 | 93.9 | 66.0 |
| Thinking | 62.0 | 48.9 | 0.0 | 88.3 | 94.7 | 81.0 |

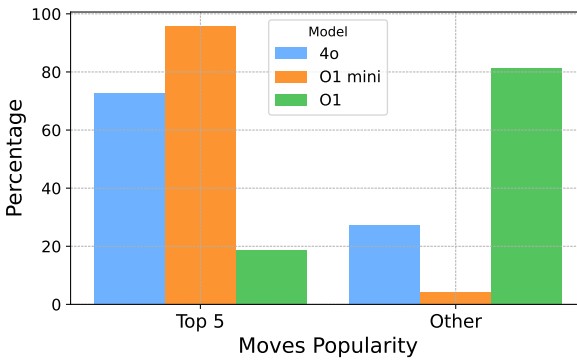

Table 5: Token count comparison of CoT, SC, ToT, and RAP on LLaMA 3.1 8B across Blocksworld, Game of 24, and GSM8K. The results highlight that scaling up inference-time techniques increases computational cost without proportionate performance gains, contrasting with the trends observed by Snell et al. (2024).

| | Blocksworld | | Game of 24 | | GSM8K | |
|---|---|---|---|---|---|---|
| | Acc ↑ | Tokens ↓ | Acc ↑ | Tokens ↓ | Acc ↑ | Tokens ↓ |
| CoT | 3.5 | $3.2 \times 10^4$ | 6.0 | $8.0 \times 10^3$ | 79.8 | $1.9 \times 10^4$ |
| SC | 4.5 | $1.7 \times 10^5$ | 6.0 | $4.0 \times 10^4$ | 86.7 | $2.4 \times 10^5$ |
| ToT | 13.9 | $1.1 \times 10^6$ | 1.0 | $3.6 \times 10^7$ | 60.0 | $4.4 \times 10^6$ |
| RAP | 46.8 | $4.9 \times 10^5$ | 1.0 | $1.5 \times 10^7$ | 87.3 | $9.9 \times 10^6$ |

Figure 3: Moves are grouped based on their popularity in online Rubik's Cube solutions. LLMs frequently generate moves commonly found in online algorithms, even though these moves maybe inaccurate.

harder tasks. This tradeoff underscores that increasing inference-time computation does not necessarily translate to proportional improvements in performance.

***Native Thinking Mode Outperforms Inference-Time Search in Large Reasoning Models.*** *While inference-time search methods such as ToT and RAP can improve performance of standard LLMs on certain tasks, these gains do not extend to LRMs (Table 4). ToT and RAP guide model reasoning through human-designed prompts to enable search imposed at inference time. LRMs, through post-training, internalize their own reasoning strategies, making such external guidance redundant. This suggests that post-training not only improves reasoning ability but also reduces the need for externally imposed inference-time search strategies.*

## 6 Conclusion

This paper examines the impact of scaling inference-time computation on improving the reasoning and planning abilities of LLMs. We show that scaling inference-time computation has limitations. We explore this by introducing Sys2Bench, a new *evaluation suite*, and conduct extensive experiments evaluating inference-time techniques across eleven diverse tasks spanning five categories, namely, arithmetic reasoning, logical reasoning, common sense reasoning, algorithmic reasoning, and planning. Our findings provide important insights into the limitations of inference-time *search* techniques.

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
