# OpenReview forum: "Inference-Time Computations for LLM Reasoning and Planning: A Benchmark and Insights"
_TMLR — Decision pending for TMLR_

### Review · Reviewer_G5yb · 2026-02-25

**Summary Of Contributions:**

The authors evaluate LLMs / early reasoning models on a collection of existing reasoning benchmarks. They compare several prompting strategies, think tree of thought, chain of thought, self-consistency.

**Audience:**

No

**Audience Explanation:**

I believe it is interesting to have a thorough comparison of the different reasoning techniques.

Unfortunately though I'll have to echo my earlier critcism that this paper is too outdated to be relevant to the TMLR readership. As I mentioned, this is an especially competitive field to publish in, and requires staying up to date to ensure the results are relevant for the target audience. Since the models are outdated, I suspect the audience will not put a lot of faith in it.

The paper could still have been useful if the dataset was particularly novel / original / a lot of effort, but as-is, it is a collection of existing (and mostly well-known!) datasets, which are already frequently evaluated on (eg HELM/MMLU contains some of these datasets, and I've certainly seen most/all datasets in previous papers). So the novelty here is missing as well.

**Broader Impact Concerns:**

-

**Claims And Evidence:**

No

**Claims Explanation:**

The paper seems overall well evaluated, on several relevant datasets. It was easy for me to follow this paper. There are also some interesting results on different prompting strategies (tree of thought, self consistency, etc).

But look, I hate to read these kinds of reviews, and I hate to write them even more myself. But I have in my career never seen a topic that is more mainstream and active than LLM reasoning, and unfortunately this just increases the competitive pressure for papers. I imagine this paper has been rejected several times, and is now resubmitted here. I say this as I believe it is just too out of date to be considered for publication at this point. It contains no references to papers since 2025, even though LRMs have really broken out and became much more practical and reliable in the last year. Eg, it contains no references to tuning LRMs with RL. And in general there has been a _lot_ of work on evaluating the reasoning capabilities of LLMs in 2025.

Furthermore, the paper evaluates on highly out-of-date models (O1 is the only actual reasoning model, which was released 1,5 years ago). This model was effectively the first 'real' (and hence experimental) reasoning model, and any improvements made on this paradigm (and there have been many) are missing. This makes it hard to have high confidence in some of the claims made (eg, that 'language models rely on retrieval rather than true understanding', which is still a hotly debated topic with rigorous arguments for but also against).

Again, I realise this project was a lot of work (and likely expensive!), and that it is not fun to have your paper become outdated because of rejections. But I'm afraid, with the current landscape of the hyper-competitive field of LLM reasoning, I cannot justify recommending accepting this paper.

**Requested Changes:**

The main changes:
- Update the literature review with recent papers (say until January 2026)
- Motivate the relevance and novelty of the research question given the new literature
- Do all evaluations for state-of-the-art reasoning models (eg at least Qwen-3 or later for open-source, and Opus4.6/ChatGPT5.2 for closed source).
- Rewrite the paper to take into account new results and insights about LRMs. Of course it's ok to have the same conclusions!
- My recommendation is to focus on evaluating different prompting strategies, which is imo the most interesting aspect of the paper.

---

> ### Author Response · Authors · 2026-04-27
> **Author Response**
>
> Thank you for your thoughtful and constructive reviews, they have significantly strengthened the paper. Below we summarize our responses to each point.
>
> **Request 1:** We have expanded the literature review with recent work in the area, adding several new references. We also extend our evaluation to reasoning models and discuss parallel studies on their evaluation.
>
> **Request 2:** We have motivated the relevance of our evaluation by extending inference-time search methods to Qwen3 reasoning models (Table 4). Specifically, we investigate whether human-guided, prompt-based inference-time search benefits recent post-trained LLMs, providing new insight into the generality of these techniques.
>
> **Request 3:** We have added results for Qwen3-8B and Qwen3-32B. Unfortunately, we were unable to extend evaluation to Claude and later GPT models, as the estimated inference cost would have exceeded \$10,000. We hope the new results nevertheless demonstrate the relevance of our work — as shown in Table 4, LRMs do not appear to benefit from prior inference-time search techniques. We conjecture this is because these techniques were human-crafted and are out-of-distribution with respect to the reasoning process LRMs internalize during post-training.
>
> **Request 4:** We have added a new paragraph in the related work section discussing reasoning models, and incorporated corresponding discussion into our updated insights section.
>
> **Request 5:** We have reframed the paper's core contribution as an evaluation suite of inference-time search methods, moving away from the benchmark framing as suggested.

---

### Review · Reviewer_iG5g · 2026-02-27

**Summary Of Contributions:**

The paper empirically investigates the impact of inference-time methods (such as chain-of-thought, tree-of-thought,...) on the performace of
LLMs. In order to do so, the authors present a benchmark dataset calles Sys2Bench, that combines several reasoning tasks from the areas of arithmetic reasoning, logical reasoning, common sense reasoning, algorithmic reasoning and planning tasks.

Based on Sys2Bench, the authors compare recent LLMs such as LLaMa, GPT and Large Reasoning Models (LRMs). They find that increasing inference-time does not necessarily increase performance.

**Audience:**

Yes

**Audience Explanation:**

Understanding the limits of different inference-time methods is of general interest, given the ever growing areas of LLM applications.

**Claims And Evidence:**

Yes

**Claims Explanation:**

The Sys2Bench benchmark appears to be comprehensive enough to enable confident experiments. Likewise, the authors compared several different LLMs to support their claims.

**Requested Changes:**

minor:
- in Table 3 appears "Algorithmic Reasoning" twice, the first appearance should be Arithmetic Reasoning.
- at the beginning of section 4 you state "... enhance System 2 abilities of LLMs". I was wondering about this phrase and whether it needs more definition.

---

> ### Author Response · Authors · 2026-04-27
> **Author Response**
>
> Thank you for the positive comments about our manuscript, we have made changes to our paper based on your review.
>
> **Request 1**: Thank you for pointing this out — we have made the suggested change.Request 1: Thank you for pointing this out — we have made the suggested change.
>
> **Request 2**: We have addressed this in the paper. The relevant paragraph in Section 4 has been updated accordingly.

---

### Review · Reviewer_ZniP · 2026-03-12

**Summary Of Contributions:**

This paper investigates inference-time computation techniques for improving LLM reasoning and planning capabilities. The authors introduce Sys2Bench, a benchmark covering 11 datasets across five task categories (arithmetic, logical, commonsense, algorithmic reasoning, and planning). They evaluate 4 inference-time methods (Chain-of-Thought, Self-Consistency, Tree-of-Thought, and Reasoning as Planning) on 7 models including both standard LLMs and large reasoning models (LRMs). The key claims are: (1) inference-time scaling has significant limitations due to inherent LLM biases, (2) tree search methods struggle with increasing task complexity, (3) LRMs still rely on pattern matching rather than true reasoning, and (4) there exists a substantial tradeoff between performance gains and computational costs.

Strengths:

1. The comparison between standard LLMs and the newer "reasoning" models (O1 series) in the context of inference-time scaling is highly relevant to the current discourse in the field.
2. The attempt to unify the evaluation of reasoning and planning under a single benchmark framework is ambitious and valuable.
3. Some interesting empirical observations about tree search limitations.

Weaknesses:

1. The benchmark is an aggregation of existing datasets (GSM8K, HotPotQA, etc.) without introducing new, challenging tasks specifically designed to test the limits of modern reasoning models.
2. Critical methods are implemented on a restricted set of models (e.g., RAP is only tested on LLaMa 3.1 8B), making general conclusions unreliable.
3. The lack of standard zero-shot baselines and the omission of recent inference-time strategies weaken the comparative analysis.
4. Substandard writing quality affecting clarity and credibility.
5. Insufficient experimental scope to support broad claims.

**Additional Comments:**

The submission addresses an important and timely topic, but the execution falls significantly below standards. The core issue is a mismatch between the breadth of claims ("inference-time computations," "comprehensive benchmark") and the narrow, outdated experimental validation.

**Audience:**

Yes

**Audience Explanation:**

The topic of inference-time computation for LLM reasoning is highly relevant to the AI research community. The benchmark concept (Sys2Bench) addresses a genuine need for standardized evaluation of reasoning methods. Some empirical observations—particularly regarding the difficulty of Rubik's Cube solving and the pattern-matching behavior of LRMs—could spark valuable discussion. However, the current execution severely limits the paper's utility. With substantial revisions addressing the methodological concerns, this work could contribute meaningfully to the field.

**Broader Impact Concerns:**

The authors provide a discussion on the computational cost trade-offs, which is a positive aspect.

**Claims And Evidence:**

No

**Claims Explanation:**

The claims made in the paper suffer from significant evidentiary gaps and logical inconsistencies:

1. The paper draws broad conclusions about the ineffectiveness of tree search methods (ToT, RAP) and the superiority of LRMs. However, the implementation of RAP is restricted exclusively to the smallest model tested (LLaMa 3.1 8B) because of technical dependencies on logits. Claiming that "tree search methods underperform" based primarily on results from an 8B parameter model, while excluding 70B and 405B models for RAP, is scientifically unsound. Without testing RAP on larger models, the authors cannot decouple the failure of the method from the limitations of the model capacity.
2. The evaluation compares CoT, SC, ToT, and RAP against each other but fails to explicitly report standard Zero-Shot baselines for all models and tasks. Without a Zero-Shot baseline, it is unclear if the complexity of methods like ToT actually provides a net positive gain over the simplest form of prompting, or if CoT is already doing the heavy lifting.
3. The paper claims to reveal that LRMs like O1 rely on retrieval/pattern matching rather than "true understanding" based on a Rubik’s Cube analysis. The metric used—frequency of aligning with "known online algorithms"—is a weak proxy for reasoning failure. An optimal reasoning process should align with known algorithms if they are correct; conversely, diverging from them might indicate hallucination rather than novel reasoning. The claim is provocative but the evidence provided is circumstantial and weakly justified.
4. The authors claim Sys2Bench is a "comprehensive benchmark." However, it consists entirely of established datasets that have been in the public domain for some time. Many of these datasets (e.g., GSM8K) are arguably "saturated" or too easy for state-of-the-art models like O1. The paper does not justify why only reasoning and planning are covered, ignoring other critical aspects of "System 2" thinking like constraint satisfaction or multi-turn strategic interaction, which limits the scope of the claimed conclusions.
5. The paper evaluates LLaMA 3.1 (8B/70B/405B) and GPT-4o/mini, which are not state-of-the-art for reasoning tasks. Critically, the authors fail to distinguish between reasoning-optimized models (e.g., O1, DeepSeek-R1, Kimi k1.5) and general-purpose models. The claim that "inference-time scaling has limitations" is misleading when tested primarily on models not designed for extensive test-time computation. The comparison between O1 (an LRM) and standard LLMs is apples-to-oranges without controlled analysis of whether similar gains can be achieved on non-reasoning models through inference-time scaling.

**Requested Changes:**

1. Implement RAP and ToT on the other models or provide a rigorous justification for why this is infeasible. Concluding that tree-search methods fail based on 8B model results alone is insufficient. If implementation on closed models (GPT) is impossible, this should be explicitly framed as a limitation rather than a general finding.
2. Include Zero-Shot,Few-Shot and nor inference-time scaling baselines (without CoT instruction) for all models and tasks. This is necessary to quantify the actual gain provided by the inference-time techniques.
3.  Address the concern regarding the "stale" dataset selection. The authors should consider including more recent, challenging datasets (e.g., MATH-500, GPQA, or newer planning domains) that are not yet saturated by models like GPT-4o or O1.
4. The writing quality requires significant improvement. The structure of the "Experiments" and "Insights" sections is repetitive, and the transition between methodology and results is abrupt. The Related Work section is sparse and fails to situate the work within the broader context of recent "System 2" reasoning literature. A thorough proofread is necessary to correct awkward phrasing and improve flow. Such as Section 2. last expressions, P9. Table...
5. The references are incomplete. The authors should include more comprehensive citations on inference-time computation, particularly recent works on "thinking time," budget forcing, and LRM evaluations that have emerged in the last 6-12 months.

---

> ### Author Response · Authors · 2026-04-27
> **Author Response**
>
> Thank you for the encouraging and constructive assessment. We are glad the core motivation and empirical observations resonate, and we appreciate the detailed feedback that has helped us substantially improve the paper.
>
> **W1: Extension of RAP to Larger Models.**
> Thank you for this suggestion. We have extended RAP to larger models, including LLaMA-70B (Tables 2 and 4). We have also added a note in the implementation details section clarifying that RAP requires access to model logits to measure action utility. While LLaMA-405B is open-source, obtaining logits at that scale was not practically feasible. We believe the extended evaluation addresses the core concern.
>
> **W2: Zero-Shot Baselines.**
> We have added zero-shot baselines to all tables. As conjectured, the results confirm that most of the performance gains are attributable to CoT.
>
> **W3: Clarification on Rubik's Cube Analysis.**
> Thank you for the feedback. We have reframed our analysis specific to Rubik's Cube. Our key observation is that LLMs tend to repeat moves commonly found in human-designed solving algorithms, but these moves are not universally applicable and are highly dependent on the current cube state. Crucially, both LLMs and reasoning models fail to identify when these moves are applicable, repeating patterns from popular algorithms without verifying whether the conditions for their use are met.
>
> **W4: Positioning of Sys2Bench.**
> We have reframed Sys2Bench as an evaluation suite rather than a benchmark, with corresponding changes throughout the paper including the abstract, introduction, and descriptions. This positions our work more accurately relative to its core focus on inference-time search prompting.
>
> **W5: Clarification on Limitations of Inference-Time Scaling.**
> We would like to clarify that our claim is not that inference-time scaling is limited in general, but rather that human-crafted prompting strategies for guiding LLM reasoning have limitations. To illustrate this, we extend popular inference-time search techniques to Qwen3 and compare against thinking and non-thinking modes. As with traditional LLMs, any gains from these techniques are inconsistent. We have added a dedicated paragraph in the Insights section discussing this finding.
>
> **W6: Dataset Selection.**
> We appreciate this suggestion. We would like to note that the majority of tasks in our evaluation suite remain far from saturated — performance ranges from close to 90% on GSM8K down to 0% on Rubik's Cube, reflecting a wide spectrum of difficulty. We believe this diversity allows readers to observe the benefits and limitations of inference-time search techniques across tasks of varying complexity, which have not been systematically ablated at this scale before.
>
> **W7: Writing Quality and Related Work.**
> We appreciate this feedback and have made significant revisions throughout the paper. We have expanded the Related Work section with recent literature, with a particular focus on reasoning models and their relation to inference-time search. We have also added a new paragraph in the introduction better motivating the need for our evaluation suite. The Insights section has been extended with additional observations pertaining to reasoning models. We have also carefully proofread the paper to address awkward phrasing, improve flow, and reduce repetition between the Experiments and Insights sections.
>
> **W8: More Updated References.**
> We have added more recent references throughout the paper, including studies on the reasoning capabilities of LRMs as recent as 2026 that highlight their limitations. These are incorporated as a new paragraph at the end of the related work section.

---

### Author Response · Authors · 2026-03-25
**Request for Extension of Response Period**

Dear Action Editor and Reviewers,

We apologize for this request. We are currently handling ICML reviews, which have required significant additional work. Combined with the substantial revisions requested by the TMLR reviewers, the overlap in deadlines makes it difficult to give both sets of reviews the careful attention they deserve.

We kindly request a 2-week extension after the ICML author response deadline (4 weeks from 03/26/2026). We appreciate your understanding.

Best,
Authors

---

> ### Comment · Action_Editor_N1AB · 2026-04-12
>
> Hello! I'm sorry for the slow response. You have been granted a 2-week extension starting from today. Please let me know if you have any follow-up questions.

---

### Decision · Action_Editor_N1AB · 2026-07-01

**Recommendation:** Accept as is

**Audience:**

Yes

**Audience Explanation:**

This paper may be of interest to those studying LLM reasoning capabilities.

**Claims And Evidence:**

Yes

**Claims Explanation:**

This paper provides a new LLM reasoning benchmark that allows the authors to investigate how different inference-time computation techniques can improve LLM reasoning and planning capabilities. The main takeaway is that CoT is often hard to beat, while more expensive tree-search methods help only in some settings and sometimes even hurt performance. The primary concern from the reviewers (especially Reviewer G5yb) was that the paper was already fairly out-of-date by the time it was submitted, but the authors took this reviewer's concerns seriously and made sufficient updates to satisfy this reviewer. I'll note a few more lingering comments from the discussion phase:
- From Reviewer iG5g: the paper could articulate the core novelty of Sys2Bench more clearly and include more details on experimental setup for the sake of reproducibility.
- From Reviewer G5yb: A downside is the lack of statistical testing, which is mentioned as an issue for evaluating reasoning models in a recent preprint: "ReasonBENCH: Benchmarking the (In)Stability of LLM Reasoning" by Nearchos Potamitis, Lars Klein, Akhil Arora.

---

> ### Author Response · Authors · 2026-07-14
>
> Thank you Dear Action Editor,
>
> We will make the additional changes and ready the paper for a camera ready release.